# Neuroprotective Effects of Betulinic Acid Hydroxamate in Intraventricular Hemorrhage-Induced Brain Damage in Immature Rats

**DOI:** 10.3390/nu14245286

**Published:** 2022-12-12

**Authors:** Aarón Del Pozo, Laura Silva, Angela Romero, María De Hoz-Rivera, María Villa, María Martínez-Vega, María Eugenia Prados, Eduardo Muñoz, José Martínez-Orgado

**Affiliations:** 1Biomedical Research Foundation, Hospital Clínico San Carlos—IdISSC, 28040 Madrid, Spain; 2Vivacell Biotechnology España S.L.U., 14014 Córdoba, Spain; 3Maimonides Biomedical Research Institute of Córdoba, University of Córdoba, 14004 Córdoba, Spain; 4Department of Cellular Biology, Physiology and Immunology, University of Córdoba, 14004 Córdoba, Spain; 5Reina Sofía University Hospital, 14004 Córdoba, Spain; 6Department of Neonatology, Hospital Clínico San Carlos—IdISSC, 28040 Madrid, Spain

**Keywords:** betulinic acid hydroxamate, intraventricular hemorrhage, neuroprotection, prematurity, rats

## Abstract

Intraventricular hemorrhage (IVH) is an important cause of long-term disability in extremely preterm infants, with no current treatment. We aimed to study in an IVH model in immature rats the neuroprotective effect of betulinic acid hydroxamate (BAH), a B55α/PP2A activator that inhibits the activity of the hypoxia-inducing factor prolyl-hydroxylase type 2. IVH was induced in 1-day-old (P1) Wistar rats by the left periventricular injection of Clostridial collagenase. Then, pups received i.p. vehicle or BAH 3 mg/kg single dose. At P6, P14 and P45, brain damage (area of damage, neurobehavioral deficits, Lactate/N-acetylaspartate ratio), white matter injury (WMI: corpus callosum atrophy and myelin basic protein signal reduction) and inflammation (TLR4, NF-κB and TNFα expression), excitotoxicity (Glutamate/N-acetylspartate) and oxidative stress (protein nitrosylation) were evaluated. BAH treatment did not reduce the volume of brain damage, but it did reduce perilesional tissue damage, preventing an IVH-induced increase in Lac/NAA. BAH restored neurobehavioral performance at P45 preventing WMI. BAH prevented an IVH-induced increase in inflammation, excitotoxicity and oxidative stress. In conclusion, in immature rats, BAH reduced IVH-induced brain damage and prevented its long-term functional consequences, preserving normal myelination in a manner related to the modulation of inflammation, excitotoxicity and oxidative stress.

## 1. Introduction

Intraventricular hemorrhage (IVH) affects about 25% of extremely low birthweight (under 1500 g) preterm newborns (ELBWN) [1,2]. IVH is a major cause of severe developmental disorders, such as cerebral palsy (CP) due to white matter injury (WMI) [1]. Blood extravasated into the brain parenchyma from disrupted germinal matrix (GM) vessels leads to brain injury firstly due to the mass effect of the hematoma and then due to the toxic effects of different components of the extravasated blood, activating harmful mechanisms such as neuroinflammation, along with excitotoxicity and oxidative stress [3,4]. Preventive anti-inflammatory treatments including prenatal steroids or prophylactic indomethacin reduce the incidence of IVH in ELBWN, but do not reduce the risk of developing CP and other complications once IVH has occurred [5]. Therefore, it is accepted that there is no current treatment for established IVH in those infants [3] and research on therapeutic strategies for this condition is warranted.

Because of their favorable toxicological profile dietary triterpenoid nutrients such as betulinic acid are important not only for the prevention of different diseases but also to develop novel derivates with improved pharmacological functionalities that can be used not only for oral delivery but also for intravenous use in life endangering conditions. Thus, the semi-synthetic triterpenoid hydroxamate derived from betulinic acid (betulinic acid hydroxamate, BAH) is an inhibitor of the prolyl 4-hydroxylase 2 (PHD2) that activates the HIF pathway [6]. Post-insult administration of BAH to hypoxic-ischemic newborn rats produces robust neuroprotective effects, in a manner related to the modulation of inflammation, excitotoxicity and oxidative stress [7].

Herein we studied if the administration of BAH post-insult would protect against brain damage induced by IVH in an immature brain. For this, we used a preclinical model that quite closely translates the typical pathophysiological and clinical events of IVH brain damage in ELBWN. In this model [8], IVH is induced by the paraventricular injection of Clostridium collagenase in 1-day-old (P1) Wistar rats, which show a brain developmental stage similar to that of preterm babies with 24-to-26 weeks of gestational age [9].

## 2. Materials and Methods

### 2.1. Animals

The experimental procedures met the European and Spanish regulations (2010/63/EU and RD 53/2013) and were designed and performed by researchers qualified in Laboratory Animal Science. Experimental protocol was approved by the San Carlos University Hospital Animal Welfare Ethics Committee (Madrid, Spain) (Protocol number: PROEX 122.4/21). FELASA recommendations were followed to preserve animal welfare and reduce suffering as well as the number of animals used.

Pregnant Wistar rats (Charles River, Barcelona, Spain) were maintained with free access to food and water. One-day-old (P1) pups were blindly assigned to sham (SHM) or collagenase-infusion groups (IVH). All groups were sex balanced within each litter. Sample size for each group was calculated based on previous experiments of our group [7,8].

Rats from both groups were similar in terms of sex distribution (male/female 21/19, 26/20 and 18/23 for SHM, IVH + Vehicle (Vehicle (VEH)and IVH+BAH, respectively, X2 = 2.97, *p* = 0.22), weight at procedure (6.2 (6.1, 6.4), 6.4 (6.2, 6.7) and 6.5 (6.4, 6.8) g for SHM, IVH+VEH and IVH+BAH, respectively, H = 5.62, *p* = 0.10) and post-procedure mortality (2/40, 5/46 and 3/41 for SHM, IVH+VEH and IVH+BAH, respectively, X2 = 1.04, *p* = 0.59).

### 2.2. IVH Induction

The experimental model has been extensively described elsewhere [8]. Briefly, IVH pups were placed prone in a stereotaxic frame (VWR International Ltd., Radnor, PA, USA) under sevoflurane anesthesia (5% induction, 3% maintenance). IVH was induced by injecting over three minutes, using a 33-gauge Hamilton syringe (HAMI65460-03, Hamilton Company, Reno, NV) attached to a syringe holder (6860, VWR International Ltd.), 0.5 µL of sterile PBS containing 0.2 U of clostridial collagenase VII-S (Sigma-Aldrich, St Louis, MO, USA). Injection site in the left germinal matrix was located using stereotactic. After 9 min, the needle was removed [8]. Then, pups were returned to their dams. Six hours after IVH, pups were randomized to receive i.p. vehicle (IVH+VEH, *n* = 46) or BAH at 3 mg/kg (IVH+BAH, *n* = 41). BAH (supplied by VivaCell Biotechnology España, S.L.U, Córdoba, Spain) was prepared in a 1 mg/mL formulation of ethanol:cremophor:saline at a ratio of 1:1:18, and further diluted in the same vehicle to administer the dose in 0.2 mL final volume. Dosage was selected after previous studies by our group demonstrating BAH neuroprotection in hypoxic-ischemic brain damage in newborn rats [7]. SHM animals (*n* = 40) were similarly manipulated but without intracerebral injection. At the end of the experiment, rats were killed by a lethal injection of thiopental sodium and fentanyl citrate. for histologic studies, rats were transcardially perfused with cold paraformaldehyde (4%) and sodium chloride (0.9%); their brains were then harvested and placed in paraformaldehyde 4%. For spectroscopy or biochemical studies, rats were perfused with sodium chloride alone and their brains snap frozen and stored at −80 °C.

### 2.3. MRI Studies

MRI was performed at P6 and P45 in a 1 Tesla benchtop MRI scanner (Icon (1T-MRI); Bruker BioSpin GmbH, Ettlingen, Germany), at the BioImaC (Universidad Complutense, Madrid, Spain), a node of the ICTS ReDiB. The technical specifications as well as the protocol to assess brain damage and Corpus callosum (CoCa) area using ImageJ 1.34 s software (NIH, Bethesda, Rockville, MD, USA) have been described elsewhere [8].

### 2.4. Neurobehavioral Studies

Neurobehavioral tests were performed at P14 or P45, as described elsewhere [8,10]. At P14, coordination (inverse geotaxis: time to turn 180° after being placed downwards on a ramp tilted at 45°) and strength (grip test: grasp reflex score after leaning a thin rod against each paw palm) were assessed. At P45, coordination (beam test: time to cross a 1 m long beam), hemiparesis (Cylinder rearing test (CRT): initial forepaw preference—left, right or both—after placing the rat in a methacrylate transparent cylinder) and memory (novel object recognition (NOR): time spent on exploration of a familiar and a novel object in a methacrylate box) were assessed. Tests were video recorded and then assessed by three different researchers blinded to the experimental group.

### 2.5. Histologic Studies

The procedure for immunohistochemistry studies in 4 µm thick brain slices, obtained at a level corresponding to plate 21 of the Paxinos and Watson Atlas [11] has been detailed elsewhere [8,12]. To assess myelin basic protein (MBP) signal at P45, MBP antibodies (1:600; Merck KGaA, Darmstadt, Germany) and corresponding Alexa-Fluor conjugated secondary antibody (1:200; Life Technologies, Madrid, Spain) were used. MBP signal intensity ratio was determined by a researcher blinded to the experimental group using the LEICA LASF Software (Leica Microsystems, Wetzlar, Germany) in microphotographs from ipsilateral and contralateral External Capsule obtained using a Leica TCS SP5 confocal microscope system (Leica, Wetzlar, Germany).

### 2.6. Biochemical and Molecular Studies

Western blot studies were performed at P6 as reported elsewhere [8]. Inflammation was studied determining the expression of Toll-like receptor 4 (TLR-4, 1:100; Santa Cruz, CA, USA), TNFα and NF-κB (both 1:100; R&D Systems, Minneapoli, MN, USA), quantified as protein measured/β-actin ratio. Oxidative stress was assessed determining protein nitrosylation using a detection kit (Oxyblot, Millipore Iberica; Madrid, Spain) according to the manufacturer’s protocol, expressed as OxyBlot/Total Lane Protein ratio. Brain samples contained 20 µg of total protein.

### 2.7. Spectroscopy Studies

Proton Nuclear Magnetic Resonance Spectroscopy (1H-NMR) was performed on frozen samples from striatal area from P6 rats at the BioImaC using a Bruker AVIII500HD 11.7 T spectrometer (Bruker BioSpin, Karlsruhe, Germany). The technical specifications have been reported elsewhere [8]. Lactate/N-acylaspartate (Lac/NAA) and glutamate/N-acylaspartate (Glu/NAA) ratios were calculated to assess brain injury and excitotoxity, respectively.

### 2.8. Statistical Analysis

Data showing a normal distribution (D’Agostino-Pearson test) were expressed as mean ± Standard Error of Mean (SEM) and compared using one-way ANOVA with the Holm–Šidack test for multiple comparisons, whereas those showing a non-normal distribution were expressed as median (IQR) and compared using Kruskall–Wallis with Dunn’s test for multiple comparisons. Contingency tables were studied using the X2 test. A *p* < 0.05 was considered significant. Statistical analysis was performed using the GraphPad Prism 9 software (GraphPad Software, San Diego, CA, USA).

## 3. Results

### 3.1. IVH-Induced Brain Damage

As observed at P6 by MRI, PVCC led to GM hemorrhage further extended to the surrounding parenchyma and ventricles, in some cases resulting in ventricular dilation (Figure 1A). Brain damage was stable over time, as observed at P45 (Figure 1A). Administration of BAH did not modify the volume of damage (Figure 1A). In the adjacent striatal area, 1H-NMR studies (Figure 1B) demonstrated increased Lac/NAA ratio after IVH. In this case, administration of BAH prevented an IVH-induced increase in Lac/NAA ratio (Figure 1B).

### 3.2. Functional Consequences of IVH

IVH resulted in impaired coordination and strength, as shown by the poorer performance in the geotaxis and grip tests (Figure 2A) at P14. IVH-induced motor impairment in the short term was prevented by BAH treatment (Figure 2A). Gross motor performance was still impaired at P45 in IVH rats. Motor impairment was still observed at P45 in IVH rats; at that time, IVH rats showed increased paresis in the contralateral forepaw as assessed using CRT, resulting in impaired coordination, with longer time needed to cross the beam in IVH than in SHM rats (Figure 2B), IVH also resulted in cognitive impairment, with IVH rats showing impaired working memory as assessed using the NOR test (Figure 2B). BAH treatment abolished IVH-induced long-term motor and cognitive impairments (Figure 2B).

### 3.3. Long Term WMI Jury after IVH

Lon-term WMI resulting from IVH was apparent at P45. Reduction in the CoCa area was demonstrable in the MRI studies (Figure 3A), whereas immunohistochemistry studies revealed a reduced MBP signal in IVH animals (Figure 3B). BAH treatment had a partial effect on IVH-induced reduction in CoCa volume. Therefore, CoCa volume in IVH+BAH animals was greater than in ICH+VEH animals but lower than in SHM animals (Figure 3A). In contrast, BAH administration fully prevented an IVH-induced decrease in MBP signal in the ipsilateral External Capsule (Figure 3B).

### 3.4. Mechanisms of Brain Injury

^1^H-NMR studies showed increased Glu/NAA values in IVH rats, corresponding with increased excitotoxicity (Figure 4A). The increase in Glu/NAA values shown in IVH-induced animals was not observed in those treated with BAH (Figure 4A). OxyBlot studies showed increased protein nitrosylation in the IVH brain, corresponding with increased oxidative stress (Figure 4B). BAH administration prevented an IVH-induced increase in oxidative stress (Figure 4B). Inflammation was increased in after IVH, as assessed using Western blot studies. Thus, increased TLR4, NF-κB and TNFα expression was detected in IVH+VEH brains (Figure 4C). BAH showed anti-inflammatory properties, with TLR4, NF-κB and TNFα expression in the brain lower than that of IVH+VEH animals and similar to that of SHM animals (Figure 4C).

## 4. Discussion

In the present work, we report that the post-insult administration of BAH resulted in neuroprotective effects in immature rats submitted to IVH induction, evaluated by neuroimaging, histological, biochemical and functional studies, in a manner linked to the modulation of inflammation, excitotoxicity and oxidative stress. There are few reports of post-insult treatments showing such neuroprotective efficacy in a very immature brain. Stem cell administration leads to reduced brain damage and preservation of myelination, by reducing inflammation and oxidative stress and promoting neuroproliferation, leading to long-term benefits in motor impairment, but this has been shown in P4 rats submitted to IVH by intraventricular injection of blood [13]. In a model similar to ours, ACTH administration resulted in reduced inflammation with reduced brain damage in rats submitted to IVH by collagenase injection at P2, but brain damage was assessed only in males, follow-up ended at P8 and did not include functional studies [14]. Our results consolidate those already reported by our group demonstrating BAH neuroprotection after hypoxic-ischemic brain damage in newborn rats [7]. The present results are even more striking considering that in the hypoxic-ischemic model, BAH is administered 30 min post-insult, and some neuroprotective effects are still observed when it is administered up to 12 h post-insult [7]. In the present study, BAH administered 6 h after IVH induction was markedly neuroprotective, pointing to a robust neuroprotective profile and a wide time window of efficacy for BAH. This interval was selected to strengthen the translational value of the model, since in preterm infants IVH is diagnosed after routine scans or when some complications arise, that is, hours or days after IVH is established [1,3]. The pathophysiology of IVH-induced brain damage is very complex because it comprises two different mechanisms of damage: one resulting from the compressive effects of the hematoma, and the other resulting from the toxic effects of the released blood products [3,4]. Although the time course of the pathophysiology of IVH-induced brain damage is not as well understood as that of hypoxic-ischemic brain damage, it is accepted that compression by the hematoma is an early process and that the toxic effects of blood products develop later [3,4]. Thus, it is conceivable that the late administration of BAH could not counteract the mass effects of hematoma and thus could not reduce the volume of damage. On the contrary, BAH could modulate the following processes initiated by toxic blood products, protecting the surrounding tissue from secondary damage. In support of this, we observed in the adjacent striatum that the IVH-induced increase in Lac/NAA value, a surrogate of brain damage [15], was not observed in IVH+BAH animals.

BAH has two complementary mechanisms that explain its neuroprotective effects. BAH is a direct PHD2 inhibitor, resulting in the stabilization and accumulation of HIF-1α, which has neuroprotective effects on acute brain damage in neonatal rodents [16,17,18]. Some of the neuroprotective effects of HIF-1α stabilization are related to the regulation of erythropoietin (EPO), VEGF and glycolytic enzyme gene transcription [17,19,20]. Furthermore, BAH activates the B55α/PP2A pathway, which in addition to inhibiting PHD2, plays an important role in vascular remodeling and induces potent anti-inflammatory effects [21,22]. Indeed, betulinic acid (BA) is a pleiotropic nutrient that mediates neuroprotection and anti-inflammatory activities by acting on other targets. Thus, it is possible that the anti-inflammatory mechanism is shared by BA and BAH. Interestingly, decarboxylated BA metabolites have been detected in human plasma and it is possible that some of these metabolites are able to mimic BAH activity [23].

All of these properties explain the pleiotropic nature of BAH neuroprotection in our experiments, with BAH modulating excitotoxicity, oxidative stress and inflammation. Blood extravasation after IVH triggers a strong inflammatory response due to brain infiltration with inflammatory cells and the effect of blood products released after hemolysis [1,3,4], resulting in the upregulation of TLR4 expression, detectable in brain tissue and inflammatory cells [24]. We observed an increased expression of TLR4 in the striatum as well as an increased expression of NF-κB and TNFα one day after IVH induction, as reported [8]. The activation of TLR4 triggers the activation of the NF-κB pathway to induce the expression of proinflammatory genes such as TNFα [24,25]. Thus, NF-κB signaling plays a key role in inflammation-based acquired immature brain damage [26]. BAH treatment abolished the IVH-induced increase in NF-κB expression. In hypoxic-ischemic newborn rats, inhibition of NF-κB signaling not only results in neuroprotection but also in neurofunctional recovery [26], as was the case in our experiments. TLR4 plays a major role in IVH-induced brain damage because of the induction of inflammation, oxidative stress and excitotoxicity [3,24,27]. Accordingly, after IVH we observed increased protein nitrosylation in brain tissue, a marker of oxidative stress seen after acute injury in the immature brain [8,28,29]. Excitotoxicity could be particularly harmful in this scenario as TLR4 activation upregulates N-methyl-D-aspartate (NMDA) signaling [27]. The complex relationship between inflammation and excitotoxicity and oxidative stress explains why pleiotropic substances such as BAH could be so effective in this condition, similar to what has been reported in newborn hypoxic-ischemic brain damage for BAH [7] and other substances [15]. Although BAH inhibition of TLR4 activation might be involved in preventing increased oxidative stress and excitotoxicity, BAH may also modulate oxidative stress and excitotoxicity through other mechanisms, such as increased EPO production by stabilization of HIF-1α or activation of the B55α/PP2A pathway [17,19,20,21,22].

Inflammation, but also excitotoxicity and oxidative stress, are particularly damaging to immature oligodendrocytes (OL) [30], which are the predominant type of OL cells in the immature brain [31]. Thus, IVH affects the survival of immature OL as well as their maturational process towards myelin-producing OL cells [4], an effect described in the model used in this work [8]. WMI in preterm infants is macroscopically detectable on MRI studies as it leads to a long-term reduction in CoCa volume, which correlates with developmental impairment [32]. Those features were reproduced in our model, as described [8], with a reduced area of CoCa observed in MRI studies at P45 in IVH rats. Consistent with their effects on inflammation, excitotoxicity and oxidative stress, BAH showed beneficial effects on the macroscopic characteristics of WMI, reducing the decrease in the CoCa area. Macroscopic WMI corresponded with histological evidence showing a decreased MBP signal in the ipsilateral External Capsule of IVH+VEH animals. The protective effect of BAH at that level was robust, with IVH+BAH animals showing an MBP signal similar to SHM animals. This is the first description of a protective effect on IVH-induced hypomyelination in very immature rat brains.

WMI is the cause of long-term motor disabilities that constitute Cerebral Palsy, as well as associated cognitive and sensory deficits [4,33]. In the model used in this work, it has been described that motor and cognitive (memory) deficiencies are still detectable when immature rats submitted to IVH become adults [8]. We corroborated these findings in IVH+VEH rats, which showed motor alterations with impaired negative geotaxis and grip test performance at P14 and hemiparesis and longer time to cross a beam at P45, as well as cognitive deficits with impairment of working memory at P45. Consistent with the protective effects shown by BAH on myelination disturbances, mid- and long-term motor and cognitive disturbances induced by IVH were not observed in animals treated with BAH.

## 5. Conclusions

In conclusion, the administration of a single dose of BAH to immature rats after IVH induction led to robust neuroprotective effects. Although BAH treatment was unable to reduce the volume of damage, it exerted some beneficial effects on perilesional tissue, reducing brain injury and protecting the myelin maturation process, thus preventing IVH-induced WMI. As a result, BAH treatment prevented the development of long-term IVH-induced motor and cognitive disabilities. All these protective effects were obtained in a manner related to the modulation of inflammation as well as excitotoxicity and oxidative stress. The remarkable protective effects of BAH on IVH-induced brain injury at biochemical, histological and functional levels, together with its pleiotropic nature point to BAH as a serious candidate to be considered for the prevention of such a devastating condition as post-hemorrhagic cerebral palsy in extremely low birthweight preterm infants.

## Figures and Tables

**Figure 1 nutrients-14-05286-f001:**
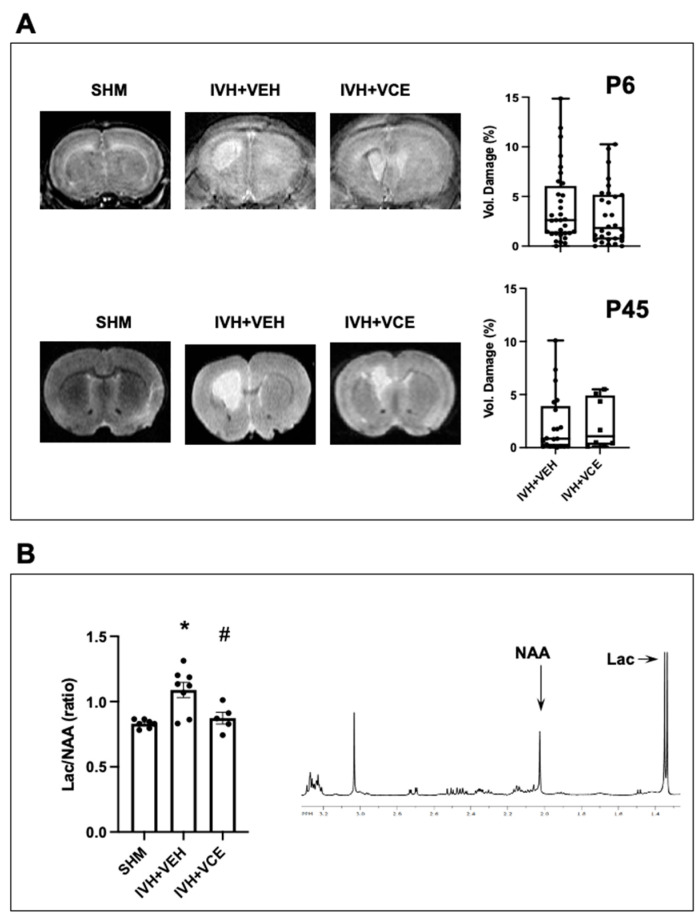
Assessment of brain damage in rats submitted to intraventricular hemorrhage (IVH) induction by paraventricular clostridium collagenase injection at day 1 (P1), then receiving vehicle (IVH+VEH) or betulinic acid hydroxamate (BAH). Non-injected pups remained as controls (SHM). (**A**). Representative T2-Weighted MRI scans obtained from SHM, IVH+VEH and IVH+BAH animals at P6 and P45, and quantification of brain damage volume. Boxes represent the median and 95% CI; whiskers represent maximum and minimum values. Mann–Whitney test: U = 405.5, *p* = 0.29. (**B**). Results from 1H magnetic resonance studies performed at P6 determining Lactate/N-acylsparate (Lac/NAA) ratio. Bars represent the mean (SEM). (*) *p* < 0.05 vs. SHM, and (#) *p* < 0.05 vs. IVH+VEH, by ANOVA with Holms–Šidack test for multiple comparisons (F(2,17) = 2.60, *p* = 0.001).

**Figure 2 nutrients-14-05286-f002:**
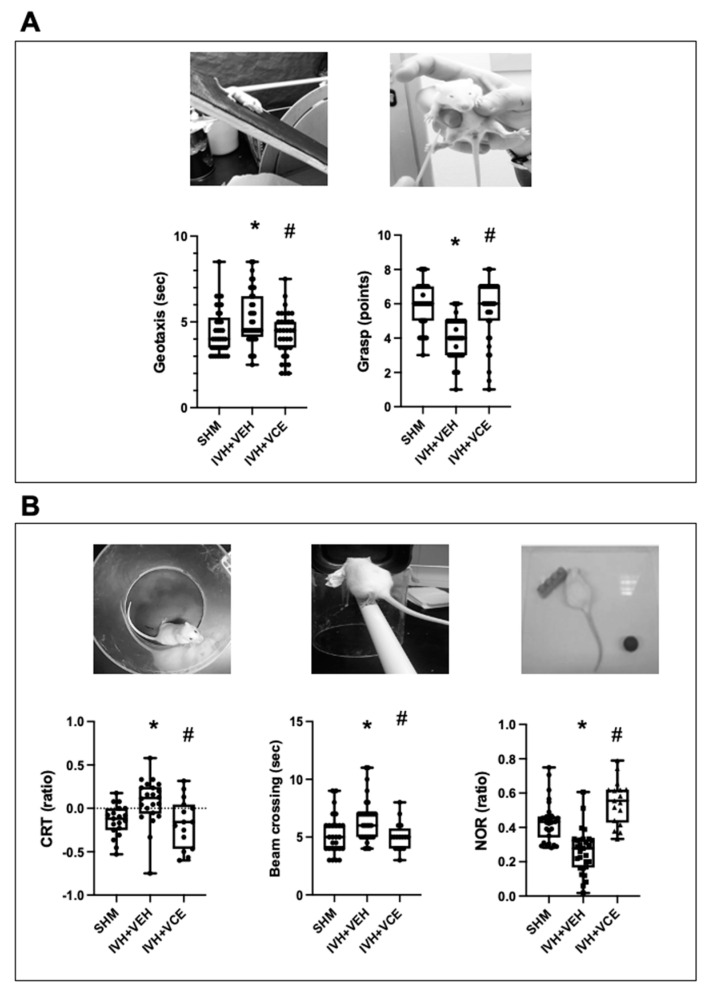
Functional consequences in rats submitted to intraventricular hemorrhage (IVH) induction by paraventricular clostridium collagenase injection at day 1 (P1), and then receiving vehicle (IVH+VEH) or betulinic acid hydroxamate (BAH). Non-injected pups remained as controls (SHM). (**A**). Neurobehavioral tests performed at P14. (**B**). Neurobehavioral tests performed at P45. Boxes represent the median and 95% CI; whiskers represent maximum and minimum values. CRT: cylinder rear test. NOR: Novel object recognition. (*) *p* < 0.05 vs. SHM, and (#) *p* < 0.05 vs. IVH+VEH, by Kruskall–Wallis with Dunn’s test for multiple comparisons (Geotaxis: H = 7.89, *p* = 0.02; Grasp test: H = 30.7, *p* < 0.0001; CRT: H = 12.24, *p* = 0.002; Beam crossing: H = 10.12, *p* = 0.006; NOR: H = 36.54, *p* < 0.0001).

**Figure 3 nutrients-14-05286-f003:**
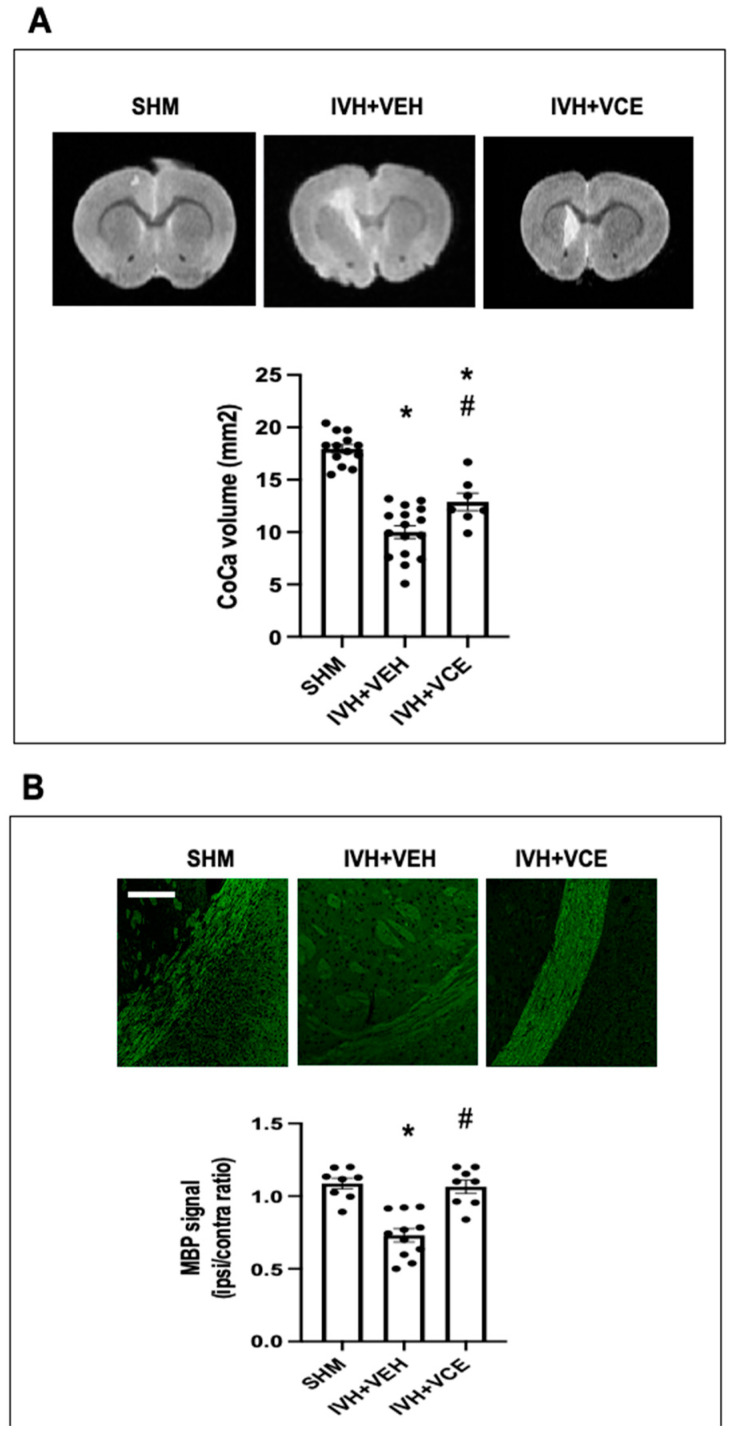
White Matter Injury in rats submitted to intraventricular hemorrhage (IVH) induction by paraventricular clostridium collagenase injection at day 1 (P1), and then receiving vehicle (IVH+VEH) or betulinic acid hydroxamate (BAH). Non-injected pups remained as controls (SHM). (**A**). Representative T2-Weighted MRI scans obtained at P45 and quantification of Corpus Callosum (CoCa) area. (**B**). Microphotographs and graphical representation of immunohistochemistry studies assessed in the ipsilateral External Capsule myelin basic protein (MBP) signal at P45. Original magnification: 200×; scale: 50 µm. Bars represent the mean (Standard Error of Mean). (*) *p* < 0.05 vs. SHM, and (#) *p* < 0.05 vs. IVH+VEH, by ANOVA with Holms−Šidack test for multiple comparisons (CoCa area: F(2,33) = 51.44, *p* < 0.0001; MBP: F(2,24) = 21.57, *p* < 0.0001).

**Figure 4 nutrients-14-05286-f004:**
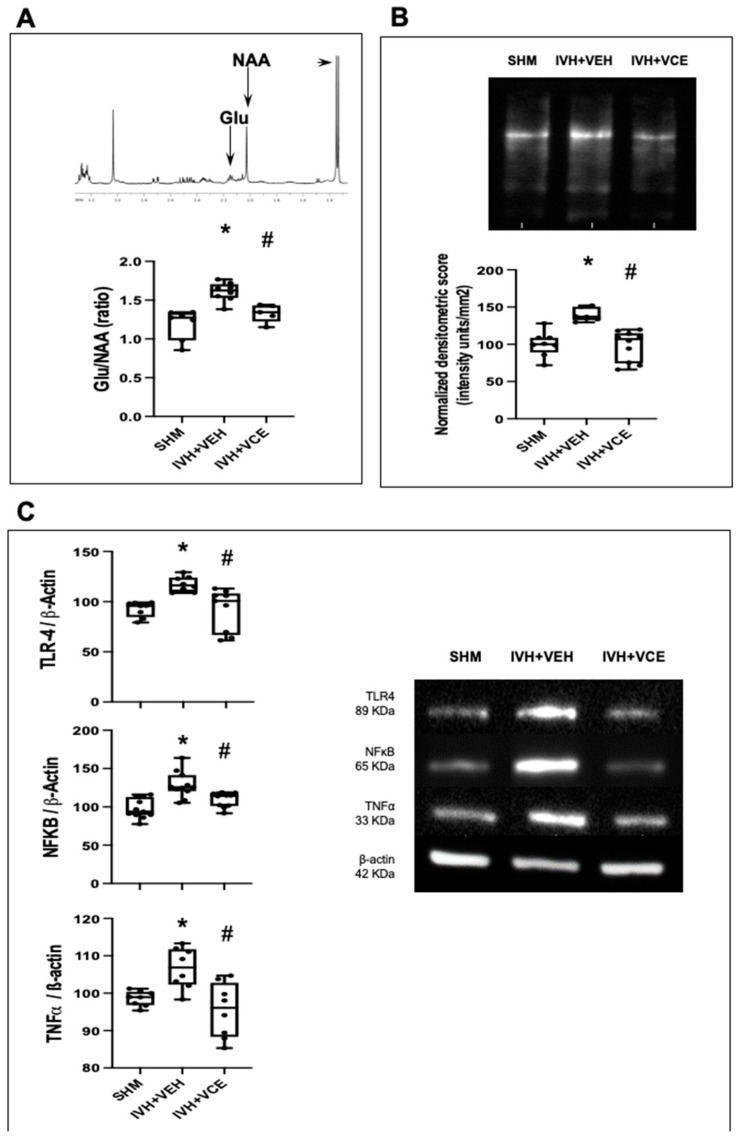
Modification of the expression of biomarkers related to excitotoxicity, oxidative stress and neuroinflammation studied at day 6 (P6) in rats submitted to intraventricular hemorrhage (IVH) induction by paraventricular clostridium collagenase injection at P1, and then receiving vehicle (IVH+VEH) or betulinic acid hydroxamate (BAH). Non-injected pups remained as controls (SHM). (**A**). Results from 1H magnetic resonance studies performed at P6 assessing excitotoxicity by Glutamate/N-acylaspartate (Lac/NAA) ratio quantification; (**B**). Representative Oxyblot film to assess protein nitrosylation and the corresponding graphical representation of the densitometric analysis; (**C**). representative samples of Western blot studies performed in brain samples and the corresponding graphical representation of the densitometric analysis. Boxes represent the median and 95% CI; whiskers represent maximum and minimum values. (*) *p* < 0.05 vs. SHM, and (#) *p* < 0.05 vs. IVH+VEH, by Kruskall–Wallis with Dunn’s test for multiple comparisons (Glu/NAA: H = 13.71, *p* = 0.001; Oxyblot: H = 13.01, *p* = 0.001; TLR4: H = 14.03, *p* = 0.0009; NF-κB: H = 17.46, *p* = 0.0002; TNFα: H = 9.96, *p* = 0.006).

## Data Availability

The datasets generated during and/or analyzed during the current study are available from the corresponding author on reasonable request.

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
