# Peer review of "Neuroprotective Effects of Betulinic Acid Hydroxamate in Intraventricular Hemorrhage-Induced Brain Damage in Immature Rats"

_nutrients, 2022, doi:10.3390/nu14245286_

Round 1

Reviewer 1 Report

Review of manuscript ref. nutrients-2060929

Title: Neuroprotective effects of betulinic acid hydroxamate in intraventricular hemorrhage-induced brain damage in immature rats.

Authors: Aaron Del Pozo , Laura Silva , Angela Romero , Maria De Hoz-Rivera , Maria Villa , Maria Martinez-Vega , Maria Eugenia Prados , Eduardo Muñoz , Jose Martinez-Orgado.

General comment: the study reports the neuroprotective effect of semi-synthetic terpenoid betulinic acid hydroxamate in an induced brain damage to newborn rats. A single administration of the drug in newborn rats submitted to intraventricular hemorrhage evoked beneficial effects on perilesional tissue, reduced brain injury and prevented the induced white matter injury. Additionally, the treatment prevented the development of long-term induced motor and cognitive disabilities. By studying different biochemical and molecular biomarkers, the authors conclude that the protective effects involved the modulation of inflammation as well as excitotoxicity and oxidative stress. The hypothesis of the study is sound and objective is comprehensive, results are clearly exposed and interesting for a future translational approach to humans. The major concern is whether the subject fits the scope of the journal Nutrients, since the compound is a semi-synthetic triterpenoid that is not found in any foodstuff and not meant to be used as a food supplement but as an intra-peritoneal injected drug. Indeed, the word nutrition, related terms or derivatives, are not mentioned in the whole text. Sincerely, this study would be better published in a pharmacology or drug therapeutic oriented journal rather than a nutritional/dietetics oriented such as Nutrients. Some specific comments are detailed below:

Specific comments:

1)      Lines 60-65; protocol reference number for the animal experimentation study should be included.

2)      Line 134; since the epigraph includes protein evaluation by western blot, perhaps the title should be Biochemical and molecular studies.

3)      Figure 4, panel C; in the picture of representative blot it should say B-actin, and this loading control should be at the bottom of the picture, not between NF-Kb and TNF-alpha.

4)      Line 140 and figure 4; there seems to be a contradiction about the target of Oxyblot assay, it says protein nitrosylation (line 140 and legend to figure) and protein carbonylation (line 225).

5)      Line 304; NMDA should be spelled out.

Author Response

- Reviewer #1:

Q1. The major concern is whether the subject fits the scope of the journal Nutrients, since the compound is a semi-synthetic triterpenoid that is not found in any foodstuff and not meant to be used as a food supplement but as an intra-peritoneal injected drug. Indeed, the word nutrition, related terms or derivatives, are not mentioned in the whole text. Sincerely, this study would be better published in a pharmacology or drug therapeutic oriented journal rather than nutritional/dietetics oriented such as Nutrients.

A1. We agree with the reviewer and we included new sentences in the text:

Introduction. Because the favorable toxicological profile dietary triterpenoid nutrients such as betulinic acid are important on only for the prevention of different diseases but also to develop novel derivates with improved pharmacological functionalities that can be used not only for oral delivery but also for intravenous use in life endangering conditions

Discussion. Indeed, betulinic acid (BA) is a pleiotropic nutrient that mediates neuroprotection and antiinflammatory activities by acting on other targets. Thus, it is possible that the antiinflammatory mechanism is shared by BA and BAH. Interestingly, decarboxylated BA metabolites have been detected in human plasma and it is possible that some of these metabolites are able to mimic BAH activity.

Q2. Some specific comments are detailed below:

Specific comments:

1)      Lines 60-65; protocol reference number for the animal experimentation study should be included.

2)      Line 134; since the epigraph includes protein evaluation by western blot, perhaps the title should be Biochemical and molecular studies.

3)      Figure 4, panel C; in the picture of representative blot it should say B-actin, and this loading control should be at the bottom of the picture, not between NF-Kb and TNF-alpha.

4)      Line 140 and figure 4; there seems to be a contradiction about the target of Oxyblot assay, it says protein nitrosylation (line 140 and legend to figure) and protein carbonylation (line 225).

5)      Line 304; NMDA should be spelled out.

A2. All these specific comments have been addressed.

Reviewer 2 Report

The manuscript by Pozo et al. aimed to study the role of betulinic acid hydroxamate (BAH) in intraventricular hemorrhage (IVH) in immature rats and if BAH administration protects the immature brain against brain damage. Importantly, the authors found a single dose of BAH in IVH damaged brain showed neuroprotective effects. These findings help the readers to understand the benefits of BAH and hold a promise to potentially apply BAH to benefit patients.

This article is an overall concise and clear report. Here list a few of my concerns that could improve the overall quality of the manuscript if addressed appropriately.

Q1: Keep all the animal information together under the Methods

Move Line 60-65 under the sub-tile 2.1 Animals (Line 66) and move Line 160-164 under 2.1 Animals as well.

Q2: Keep bold highlights for the entire first sentence of each figure legends and the alphabetical sub-titles too, such as A), B).

Q3: For the Discussion section, it would be great if the authors can present and resolve some conflicting findings regarding the benefits of BAH in IVH brains, in addition to the consistent findings related to this manuscript (Line 256).

Author Response

  • Reviewer #2.

Q1: Keep all the animal information together under the Methods

Move Line 60-65 under the sub-tile 2.1 Animals (Line 66) and move Line 160-164 under 2.1 Animals as well.

Q2: Keep bold highlights for the entire first sentence of each figure legends and the alphabetical sub-titles too, such as A), B).

Q3: For the Discussion section, it would be great if the authors can present and resolve some conflicting findings regarding the benefits of BAH in IVH brains, in addition to the consistent findings related to this manuscript (Line 256).

The article has been revised and the specific comments of the reviewer have been addressed.

Round 2

Reviewer 1 Report

The authors have conveniently addressed my comments and queries